# Formulation of Creams Containing *Spirulina Platensis* Powder with Different Nonionic Surfactants for the Treatment of Acne Vulgaris

**DOI:** 10.3390/molecules25204856

**Published:** 2020-10-21

**Authors:** Liza Józsa, Zoltán Ujhelyi, Gábor Vasvári, Dávid Sinka, Dániel Nemes, Ferenc Fenyvesi, Judit Váradi, Miklós Vecsernyés, Judit Szabó, Gergő Kalló, Gábor Vasas, Ildikó Bácskay, Pálma Fehér

**Affiliations:** 1Department of Pharmaceutical Technology, Faculty of Pharmacy, University of Debrecen, Nagyerdei körút 98, 4032 Debrecen, Hungary; jozsa.liza@pharm.unideb.hu (L.J.); ujhelyi.zoltan@pharm.unideb.hu (Z.U.); vasvari.gabor@pharm.unideb.hu (G.V.); sinka.david@pharm.unideb.hu (D.S.); nemes.daniel@pharm.unideb.hu (D.N.); fenyvesi.ferenc@pharm.unideb.hu (F.F.); varadi.judit@pharm.unideb.hu (J.V.); vecsernyes.miklos@pharm.unideb.hu (M.V.); bacskay.ildiko@pharm.unideb.hu (I.B.); 2Doctoral School of Pharmaceutical Sciences, University of Debrecen, Nagyerdei körút 98, 4032 Debrecen, Hungary; szabjud@med.unideb.hu (J.S.); vasas.gabor@science.unideb.hu (G.V.); 3Department of Medical Microbiology, Faculty of Medicine, University of Debrecen, Nagyerdei körút 98, 4032 Debrecen, Hungary; 4Department of Biochemistry and Molecular Biology, Faculty of Medicine, University of Debrecen, Egyetem tér 1, 4025 Debrecen, Hungary; kallo.gergo@med.unideb.hu; 5Department of Botany, Institute of Biology and Ecology, Faculty of Science and Technology, University of Debrecen, Egyetem tér 1., 4032 Debrecen, Hungary

**Keywords:** *Spirulina platensis*, nonionic surfactants, acne vulgaris, transcutol, antioxidant, antimicrobial, topical formulations, cutibacterium acnes, UVB-induced oxidative stress

## Abstract

Natural products used in the treatment of acne vulgaris may be promising alternative therapies with fewer side effects and without antibiotic resistance. The objective of this study was to formulate creams containing *Spirulina (Arthrospira) platensis* to be used in acne therapy. *Spirulina platensis* belongs to the group of micro algae and contains valuable active ingredients. The aim was to select the appropriate nonionic surfactants for the formulations in order to enhance the diffusion of the active substance and to certify the antioxidant and antibacterial activity of *Spirulina platensis*-containing creams. Lyophilized *Spirulina platensis* powder (SPP) was dissolved in Transcutol HP (TC) and different types of nonionic surfactants (Polysorbate 60 (P60), Cremophor A6:A25 (CR) (1:1), Tefose 63 (TFS), or sucrose ester SP 70 (SP70)) were incorporated in creams as emulsifying agents. The drug release was evaluated by the Franz diffusion method and biocompatibility was tested on HaCaT cells. In vitro antioxidant assays were also performed, and superoxide dismutase (SOD) and 2,2-diphenyl-1-picrylhydrazyl (DPPH) assays were executed. Antimicrobial activities of the selected compositions were checked against *Staphylococcus aureus* (*S. aureus*) and *Cutibacterium*
*acnes* (*C. acnes*) (formerly *Propionibacterium acnes*) with the broth microdilution method. Formulations containing SP 70 surfactant with TC showed the most favorable dissolution profiles and were found to be nontoxic. This composition also showed significant increase in free radical scavenger activity compared to the blank sample and the highest SOD enzyme activity was also detected after treatment with the cream samples. In antibacterial studies, significant differences were observed between the treated and control groups after an incubation time of 6 h.

## 1. Introduction

Acne vulgaris is a multifactorial chronic inflammatory skin disease involving pilosebaceous units and mainly affects adolescents. It has four main pathogenetic mechanisms: Increased sebum production, follicular hyperkeratinization, bacterial hypercolonization, and inflammation [1].

*Cutibacterium acnes (C. acnes)* and *Staphylococcus aureus (S. aureus)* bacteria are the most common causes of skin inflammation and the formation of acne vulgaris [2]. The colonization of *C. acnes* in the ducts of the sebaceous follicles breaks down sebum to be consumed as food. As the number of bacteria increase, inflammation occurs, which results in an immune response, but *S. aureus* is also responsible for localized cutaneous infections [3].

In acne, reactive oxygen species (ROS) may be released from the impacted, damaged follicular walls and can influence the progress of inflammation. Therefore, antioxidants are beneficial for facilitating the repair of the damage caused by these free radicals; thus, they can be used in acne therapy [1,4,5].

Topical therapy is essential in the treatment of mild to moderate acne vulgaris [6]. However, local antibiotic therapy is associated with several side effects such as skin irritation and bacterial resistance when used for a long period [7]. Therefore, there is a need to develop new natural alternative medications for treating acne.

*Arthrospira* (*Spirulina*) is a genus of prokaryotic filamentous cyanobacteria that belongs to the group of microalgae. The most commonly used species are *Arthrospira maxima (A. maxima)* and *Arthrospira platensis (A. platensis)*. These species possess diverse biological and nutritional significance and are often used as dietary supplements because of their high protein content (approximately 60–70%). Besides protein, the main bioactive compounds of *Arthrospira platensis* are phycocyanin, B-group vitamins (vitamins B12, A, E, and D), numerous microelements (e.g., Ca, Fe, P, I, Mg, Zn, Se, Cu, Mn, Cr, K, and Na), unsaturated fatty acids (e.g., gamma-linolenic acid), beta-carotene, and superoxide dismutase (SOD) enzyme [8,9]. Phycocyanin is a natural, protein-bound, blue-colored pigment composed of α and β polypeptide subunits [10]. This natural product has the ability to scavenge free radicals, thus avoiding oxidative damage and preventing premature apoptosis of cells [8]. Based on the literature, it improves skin elasticity and slows down skin aging due to its antioxidant effect [11,12]. It has also been reported that phycocyanin can favorably modulate apoptotic pathways and promote wound healing in skin cells exposed to UV radiation [8]. SOD is a metal-containing enzyme found in large amounts in algal species. This enzyme is able to neutralize superoxide anion radicals by converting them to hydrogen peroxide (H_2_O_2_) and oxygen (O_2_) [13]. Antibacterial activity of *Spirulina (Arthrospira) platensis* has also been described [14]. However, preformulation studies and appropriate excipients are inevitable in order to develop creams with optimal drug release and skin penetration [15].

Besides active substances, excipients can also cause local cutaneous irritation that may lead to noncompliance and poor outcomes during therapy. It is crucial to choose the appropriate ingredients when making a topical formulation [16,17]. The selection of surfactant in topical formulations should be based on its efficacy in enhancing skin permeation, its physicochemical and biological compatibility with the other components, and its dermal toxicity [18,19]. Surfactants may cause local irritation, membrane damage, and cell death under certain concentrations. In order to avoid this, in vitro cytotoxicity tests for topical dosage forms are required [20]. Nonionic amphiphilic surfactants are considered safe and do not cause skin irritation when used at the appropriate concentration [21]; however, the cytotoxicity of these surfactants mainly depends on their structure [20].

This study was conducted to formulate creams containing natural active substances with biocompatible tensides that act against acne vulgaris. *Spirulina platensis* powder (SPP) with different types of nonionic surfactants (Tefose 63 (TFS), sucrose ester SP 70 (SP70), Polysorbate 60 (P60), and Cremophor A6:A25 (CR)) was used in our preparations. To enhance the permeability of SPP and to improve its solubility, TC was also added to the formulations [22]. The Franz diffusion method was used to model the release of the active substance (phycocyanin) from the vehicle and to investigate its membrane diffusion. Diffusion coefficients and release rates were also determined. These parameters refer to the kinetic profile of phycocyanin permeation through the skin. In order to certify the safety profiles of the developed formulations and their excipients, 2-(4,5-dimethyl-2-thiazolyl)-3,5-diphenyl-2H-tetrazolium bromide (MTT) viability test on HaCaT cells was also performed. In the antioxidant assay, UV-B radiation was exposed to HaCaT keratinocytes in order to generate ROS and the activity of the SOD enzyme was measured to demonstrate the beneficial antioxidant effect of the formulations. Free radical scavenging activity of formulations containing TFS and SP 70 was also examined. The antibacterial activity of creams was investigated on *C. acnes* and *S. aureus* [2].

## 2. Results

### 2.1. Macroscopic Properties and pH Measurement

The compositions of formulated creams are shown in Table 1.

The evaluation of the macroscopic characteristics of the formulations prepared was performed immediately after preparation. Compositions V–VIII had a homogeneous, pale bluish-green appearance. The formulations I–IV were heterogeneous, as they contained SPP in suspended form. The pH value of each formulation ranged from 6.5 to 7.0 (Table 2). The normal skin surface pH is around 4.7; however, topical formulations that have the pH range of 4–7 are accepted [23]. Although preparations with a more acidic pH are better for the skin, it could influence the results of antioxidant and antibacterial tests in our study. That is why neutral pH was chosen. Variations in pH levels may affect the radical scavenging activity of *Spirulina platensis*, which is the best between pH 8.0 and 9.5 [24].

### 2.2. Texture Analysis

One of the most important quantitative methods for analyzing the mechanical properties of a cream is texture analysis [25,26]. Compression tests were evaluated with different cream compositions (Table 1).

The compression forces (N) required to insert the probe to a given distance into formulations I–VIII are shown in Figure 1. The maximum value of compression force denotes the firmness of the formulation [25]. According to the investigation, different compositions need different amounts of force. Formulations containing SPP in suspended and dissolved forms were compared with each other. The results showed that creams containing SPP in dissolved form with TC demonstrated lower resistance to deformation compared to those formulations with the same emulsifier where SPP was in suspended form, as seen in Figure 1. Significant (*p* < 0.05) differences were detected between compositions I and IV, II and VI, III and VII, and IV and VIII. A lower resistance value indicates a lower level of firmness, which is a desirable factor for spreading on the skin. Higher compression values indicate hard cream consistency that may hinder the liberation of active agent [27].

### 2.3. In Vitro Diffusion Studies

Figure 2 shows the cumulative amount of phycocyanin diffused from the different formulations (%) against time (minute). Each composition contained 5 g of SPP, which is equivalent to 23.07 ± 0.8 mg phycocyanin [28].

According to our experiments, compositions in which SPP was previously dissolved in TC demonstrated higher diffusion values compared with suspended formulations using the same emulgent. The release of phycocyanin from the different compositions after 105 min can be sorted based on the following descending order: VII ˃ VIII ˃ III ˃ IV ˃ VI ˃ II ˃ V ˃ I.

Significant differences were detected between the diffusion of compositions II and VI, III and VII, and IV and VIII at *p* < 0.05 examined at the same times (after 15, 30, 45, 60, 75, 90, and 105 min). As expected, with the addition of TC to the creams, greater release of phycocyanin was observed in case of compositions V, VI, VII, and VIII. Our results show that the drug release was better from compositions containing TFS or SP 70 as the emulsifying agent, even without using TC, compared with the compositions containing CR and P 60 emulsifiers.

The highest diffused amount of the active substance was achieved from formulations VII and VIII, in which SPP was dissolved in TC and the emulsifying agents were SP 70 and TFS. The maximum diffused amount of phycocyanin reached 39.81 ± 0.83% (9.18 ± 0.38 mg) in the case of composition VII and 39.80 ± 1.15% (9.18 ± 0.41 mg) for composition VIII. It was determined that the difference between the diffused phycocyanin amount from these two compositions was not statistically significant (*p* < 0.05) at the end of the experiment. Creams containing P60 as the emulsifier showed the lowest amount of penetrated drug and this did not increase significantly with the addition of TC to the formulation.

The diffusion profiles of the dissolved and suspended compositions with the same emulsifying agents were compared with each other. The calculated similarity factors are shown in Table 3. Two preparations are considered similar if their similarity factor (*f2* value) is between 50 and 100.

According to the calculated *f*_2_ similarity factor values, a high-degree similarity was confirmed when the dissolution profiles of composition V and composition I were compared with each other. Interestingly, compositions VI and II can also be considered similar, since the *f*_2_ value (53.91) was between 50 and 100 [29].

The phycocyanin release rate (k) was estimated from the slope of the amount of drug released per unit area versus the square root of time. The diffusion coefficient (D) of the drug was determined from the amount of drug released per unit area, the initial concentration, and the diffusion time. Release rates and diffusion coefficient values are listed in Table 4 [30,31]. Significant differences (*p* < 0.05) can be seen between the diffusion coefficient values of compositions III and VII, IV and VIII, and II and VI. In these cases, statistically significant lower results were detected for the suspended forms in comparison with the composition containing dissolved SPP.

### 2.4. MTT Viability Assays on the HaCaT Cell Line

#### 2.4.1. Cytotoxicity of SPP and Excipients

MTT cytotoxicity tests were carried out on HaCaT cell monolayer. The cytotoxicity of 0.50 (*m*/*m*) % solutions of nonionic surfactants, TC and SPP, was examined. The excipients and the SPP were dissolved in PBS. As it can be seen on Figure 3, the type of emulsifier agent greatly affects cell viability. The most significant decrease was detected in the case of P 60 and CR emulgents. The cell viability was less than 50% compared to the negative control (Phosphate Buffered Saline (PBS)) group. SP 70 had the highest cell viability (93.05 ± 2.20%). According to the results, it was found that SP 70 and TFS were well tolerated. There were significant differences between TC-treated and PBS-treated (negative control) groups. TC reduced cell viability to 70.46 ± 4.37%. SPP dissolved in PBS reduced cell viability by only 5.5%, and so it was found to be well tolerated.

#### 2.4.2. Cytotoxicity of Formulated Compositions

MTT cytotoxicity assays were performed on HaCaT cell monolayer with each sample prepared with the Franz diffusion cells using PBS as receptor phase. Test solutions of 1 mL were taken at the 105th minute, when the diffusion of phycocyanin reached its maximum. Creams with the same compositions but without SPP were also tested. The results of the experiments are shown in Figure 4.

Creams without TC demonstrated higher cell viability values than compositions containing TC with the same emulsifier. Formulations containing different emulsifiers showed different degrees of toxicity. Among the creams, formulations containing the sucrose ester SP 70 surfactant were less toxic. The cell viability was 87.24% ± 1.0% in the case of composition IV and 83.15% ± 0.82% in the case of composition VIII. Compositions I, V, II, and VI showed the lowest cell viabilities. So creams with Polysorbate 60 and Cremophor were shown to have the most toxic effects on HaCaT cells. In the case of preparations without SPP, there was no significant difference in cytotoxicity compared to SPP-containing creams. Based on these results, it was found that cream containing SPP with SP 70 and TC is well tolerated, as it reduced cell viability by only 13%.

### 2.5. In Vitro Antioxidant Activity Tests

#### 2.5.1. Determination of Superoxide Dismutase Activity on HaCaT Cell Line

Formulations containing TFS and SP70 as emulsifying agents and SPP in suspended (III, IV) and dissolved forms (VII, VIII) were selected for the in vivo antioxidant experiment according to the results of drug release studies and MTT cytotoxicity tests.

The SOD enzyme activity of the non-UV group was taken as 100%. The SOD activities of the treated groups were compared with the enzyme activities of the controls, which were pre- or posttreated with compositions VII and VIII without SPP. As positive control (±)-6-hydroxy-2,5,7,8-tetramethylchromane-2-carboxylic acid (Trolox) was used in pretreatment. Trolox is a vitamin E analog, which is widely used in antioxidant studies as positive control. Figure 5 demonstrates that in the group exposed to ultraviolet radiation-B (UVB), where no pretreatment or posttreatment of cream was used, the level of SOD enzyme was significantly reduced compared to the non-UV group. The drastically decreased level of antioxidant enzyme activity in HaCaT cells may have been due to the high degree of oxidative stress caused by UVB radiation, which may have led to severe cell damage. During the pretreatment we observed a smaller decrease in SOD values. The best result was obtained with the pretreatment of composition VIII where SPP was dissolved in TC formulated with SP 70 surfactant (see Figure 5b). In this case, the mean percentage of SOD activity after UVB irradiation was 26.45 ± 0.85%. The posttreatment could not prevent a significant decrease in the enzyme level caused by UVB radiation.

#### 2.5.2. Determination of 2,2-Diphenyl-1-picrylhydrazyl (DPPH) Radical Scavenging Activity

The percentage of antioxidant activity (AA%) of the formulations III, IV, VII, and VIII with or without SPP was determined based on DPPH activity assay. The DPPH assay showed that the antioxidant activity of *Spirulina platensis* extract (10 mg/mL) was approximately 68% of ascorbic acid (Figure 6). In comparison, the antioxidant activity of the SPP-containing compositions were significantly higher than the activity of the same compositions but without SPP. Creams without SPP did not show significant radical scavenging activity.

According to our results, compositions VII and VIII were able to significantly inhibit DPPH mean oxidation. Composition VIII, which contained sucrose ester SP 70 and TC, showed the most effective antioxidant activity in the DPPH assay. Formulations containing the SPP in suspended form (comp. III and IV) showed less activity than those containing the active ingredient in dissolved form.

The EC_50_ (efficient concentration) values of Spirulina platensis and ascorbic acid were also calculated from the results. Ascorbic acid had EC_50_ value of 0.015 mg/mL while *Spirulina platensis* had slightly higher EC_50_ value. According to our measurements, 0.894 mg/mL *Spirulina platensis* can cause 50% reduction in the DPPH concentration.

### 2.6. Antibacterial Test

The broth microdilution method was used to estimate the antibacterial effect of compositions VII and VIII. The antibacterial tests were carried out for 6 h on *S. aureus* (Figure 7a) and *C. acnes* (Figure 7b) species. Aknemycin™ ointment (20 mg/g erythromycin) was used as positive control. The negative control group was treated with only PBS in both cases. The effect of *Spirulina platensis* was also tested separately in the form of a 0.25% solution made with PBS.

The compositions showed less activity against *S. aureus* bacterium after two hours of incubation (2 h) compared with the activity measured against *C. acnes*. After six hours of incubation (6 h), the effectiveness of the formulations increased in both cases against *S. aureus,* and a greater increase was detected compared with the 2-h measurement. Overall, *C. acnes* was found to be more sensitive to the treatments, according to our results. Figure 7b shows that formulation containing sucrose stearate SP 70 (comp. VIII) was able to approach the effect of Aknemycin™, as the treatment with this formulation significantly decreased *C. acnes* viability to 65.34 ± 1.76%. Significant differences were observed between the treated and the control group after 2-h and 6-h incubation times, too, in the case of *C. acnes*. Treatment with a formulation containing sucrose stearate SP 70 emulgent was more effective in both cases.

*S. aureus* was less susceptible to the treatments, according to our experiment. However, there were also statistically significant differences between the viability of the negative control (PBS) and the treated groups after 6 h of incubation, as shown in Figure 7a. In this case, composition VIII was found to be more effective again, as it decreased S. aureus viability to 72.59 ± 1.98%.

Treatment only with the *Spirulina platensis* solution proved to be effective against both types of bacteria. In the case of *S. aureus,* cell viability decreased to 65.49 ± 0.97%, while in the case of *C. acnes,* it decreased to 63.67 ± 1.88%. According to our examinations, compositions without SPP did not influence the viability of bacteria significantly. Therefore, it was concluded that the excipients alone do not have antibacterial effect.

## 3. Discussion

Products containing natural active substances have an increasing role in the treatment of acne vulgaris. The external application of *Spirulina platensis* is not so widespread, although Spirulina contains valuable cosmetic ingredients. Based on the literature, it reduces dryness and itching of the skin, inhibits the formation of acne, moderates irritation, and stimulates metabolism at the cellular level [9,12].

In the present study, eight different topical formulations containing lyophilized *Spirulina platensis* extract were prepared. The use of suitable excipients and surfactants is essential in the formulation of external pharmaceuticals as they can enhance the penetration of the active ingredient through the barriers [16,18]. According to our previous experiments, TC as solubilizing agent together with SP 70 emulsifying agent enhanced the penetration of *Silybum marianum* extract in a topical dosage form [19].

In the first part of our experiments, formulations with different surfactants were analyzed for consistency, dissolution profile, and biocompatibility. Formulations that contained TC as a penetration enhancer with sucrose ester SP 70 and TFS emulsifying agent showed better drug release and consistency. The maximum diffused amount of phycocyanin (at 105 min) reached 39.81 ± 1.01% in the case of composition containing TFS and 39.80 ± 4.00% for composition containing SP 70. TC is a good solubilizing agent that may affect the skin penetration and bioavailability of SPP in the formulations. It is also reported to be nontoxic and biocompatible with the skin [32,33,34]. Surface active agents may influence the dissolution and permeability of active pharmaceutical ingredients (API) [35,36,37]. Csizmazia et al. reported that sucrose esters can increase the skin penetration and permeation of ibuprofen efficiently [38]. They are natural and biodegradable excipients with well-known emulsifying, solubilizing, and penetration-enhancing behaviors. Besides, they can improve the release, distribution, and bioavailability of active ingredients [37,39,40]. In vitro release studies carried out by us showed that surfactants influenced the dissolution of phycocyanin from the carrier system. In our study, formulations containing TFS and sucrose ester SP 70-type surfactant together with TC increased drug solubility and drug permeability, resulting in a favorable diffusion profile. Also, these formulations showed optimal consistency values according to the texture analysis studies. The nonionic surfactant TFS is a physical mixture of three components: Polyethylene glycol (PEG)-6 palmitostearate, ethylene glycol palmitostearate, and PEG-32 palmitostearate. It gives creams a pleasant texture with good dissolution of API [41,42].

In order to prove the safety of compositions, in vitro cell viability assays were performed. Cytotoxicity investigations are essential to determine the safety profiles of topical formulations. HaCaT cells are immortalized human keratinocytes from adult skin and were derived from a 62-year-old man suffering from melanoma [43]. The HaCaT cell line has been proposed as an in vitro model for the study of keratinocyte function and for screening the biocompatibility of topical formulations [44,45]. This cell line is also suitable for the assessment of the skin irritancy potential of excipients [46]. Early identification of cytotoxicity and the irritating effect of ingredients can be detected by in vitro cytotoxicity assays [47]. In our study, MTT viability tests were performed on HaCaT cell line. The cytotoxicity assay showed that compositions with the surfactants sucrose ester SP 70 and Tefose were more tolerable than the compositions with P 60 and CR. According to in vitro cytotoxicity tests on HaCaT cells, the compositions with SP 70 were less toxic; the cell viability was 87.24% ± 1.0% in the case of composition IV and 83.15% ± 0.82% in the case of composition VIII.

According to our preformulation study data (in vitro dissolution test, texture analysis study, MTT test), compositions III, IV, VII, and VIII with TFS and SP 70 surfactants were selected for further antimicrobial and antioxidant assays.

*C. acnes* can evoke local inflammation by producing neutrophil chemotactic factors, and the attracted neutrophils release inflammatory mediators such as ROS in the inflammatory tissue. Removal of the ROS can significantly reduce cell damage that may occur during acne inflammation [48]. It has already been proved that a formulation with an antioxidant effect as well as an antimicrobial effect could be more effective for acne vulgaris therapy [4].

UV radiation can lead to oxidative stress in the keratinocytes of the skin and it activates several signaling pathways. On exposure to UV radiation, the generation of ROS increases in the keratinocytes [49,50]. In our experiments, the large extent of oxidative stress caused by UVB radiation led to severe cell damage. The oxidative damage may have been the reason why the level of antioxidant enzyme in HaCaT cells is so drastically reduced. SOD, one of the main antioxidant enzymes, neutralizes free radicals generated during oxidative stress, thus preventing cell damage [51,52]. The antioxidant effect of *Spirulina platensis* is mainly due to its phycocyanin and SOD enzyme content, which both have the ability to scavenge free radicals [53,54]. That is why recent trends suggest that microalgae are a promising group in the cosmetics industry and contain valuable active substances such as antioxidants and pigments [55,56]. Gunes et al. stated that skin creams containing 1.125% *Spirulina platensis* extract exert wound healing and antioxidant activities on keratinocytes [55].

In the in vitro antioxidant test on the HaCaT cell line, dissolved SPP in TC (VII, VIII) with Tefose and SP 70 surfactant caused a significantly higher SOD level. Our results demonstrate that the effectiveness of SPP was better when it was dissolved in TC compared with compositions (III and IV) where SPP was in suspended form. However, the type of surfactant also influenced the antioxidant effect of creams. Pretreatment with composition VIII containing the sucrose ester SP 70 emulgent resulted in the lowest reduction in SOD activity caused by UVB radiation.

The non-enzymatic antioxidant scavenging activity of Spirulina was also measured. DPPH free-radical scavenging assay was performed. Wan-Loy Chu et al. reported that the radical scavenging activity of the S. platensis extract was at least 50% of vitamin C and vitamin E [57]. In our studies, we found that the radical scavenging activity of the 10 mg/mL alcoholic solution of *Spirulina platensis* was approximately 68% of ascorbic acid. Compositions containing sucrose ester SP 70 and TFS in dissolved form showed significantly higher antioxidant activity than composition III and IV.

Topical antibiotic treatment is usually the first choice in acne therapy. However, the increase in bacterial resistance has led to recommendations against the use of topical antibiotics as monotherapies, and combined therapies with other topical agents are suggested [58,59,60].

In acne vulgaris, *C. acnes* and *S. aureus* bacteria are responsible for skin inflammation, as it was reported by Weber et al. [2]. *C. acnes* produces enzymes that lead to subsequent inflammatory reactions in the surrounding dermis during the occurrence of acne [48]. In acne lesions, the concentration of *S. aureus* also increases and can cause inflammatory skin disease with pustules through the release of extracellular toxins and enzymes [2].

Fanelli et al. reported that 43% of participants were colonized with the bacteria *C. acnes* in one cross-sectional study of patients who were undergoing evaluation for acne [3].

It is essential to avoid *C. acnes* overgrowth in acne therapy [61]. However, the widespread and long-term use of antibiotics in the treatment of acne has resulted in the spread of resistant strains of *C. acnes*; therefore, natural compounds containing antimicrobial activity are gaining attention [62]. El-sheekh et al. stated that *S. platensis* has antibacterial activity against Gram-positive and Gram-negative bacteria [63]. According to Mala et al., the acetone extract of *Spirulina platensis* showed moderate antimicrobial activity against *S. aureus* [64]. The aqueous extract of *Spirulina platensis* showed an MIC (Minimum Inhibitory Concentration) value of 1.5 ± 0.1 mg/mL against *C. acnes* [65].

In the antibacterial examination, our formulations were tested on both *S. aureus* and *C. acnes* bacteria. The exact mechanism of action of *Spirulina platensis* is unknown, but there are papers describing some possible pathways. The antimicrobial activity of *Spirulina platensis* could be related to the synergistic effect of its fatty acid components and to its polysaccharide content [12]. It was also hypothesized that fatty acids kill microorganisms by leading to the disruption of the cellular membrane [63,65,66].

Against *C. acnes*, clindamycin and erythromycin are the most common antibiotics [67,68]; hence, Aknemycin cream containing erythromycin was used as positive control in the antibacterial test. The results showed that SPP-containing cream compositions significantly reduced the viability of *C. acnes* and *S. aureus* compared with the negative control (PBS) group. Creams with sucrose ester SP 70 surfactant decreased bacteria viability more than those with TFS after 6 h of incubation. These results also show that the type of surfactant influenced the bioavailability and the antimicrobial activity of SPP.

Our work points out the differences between preparations with various nonionic surface-active agents. The surfactants influenced the consistency of creams and the release of SPP as well as the antioxidant and antibacterial efficacy of our formulations.

## 4. Materials and Methods

### 4.1. Materials

Polysorbate 60 [2-[(2R)-2-[(2R,3S,4R)-3,4-bis(2-hydroxyethoxy)oxolan-2-yl] -2- (2-hydroxyethoxy)ethoxy]ethyl octadecanoate (Chemical Abstracts Service (CAS) Number: 9005-67-8)] was obtained from Sigma-Aldrich Buchs (St. Gallen, Switzerland). Transcutol HP [2-(2-ethoxyethoxy)ethanol (CAS Number: 111-90-0)] and Tefose 63 (CAS Number: 9004-99-3) were a kind gift from Gattefossé (Lyon, France). Cremophor A6 [Polyethylene glycol 260 mono(hexadecyl/octadecyl) ether (CAS Number: 85941-44-2)] and A25 [Macrogol (25)-cetostearyl ether, (CAS Number: 68439-49-6)] was obtained from BASF Company (Ludwigshafen, Germany). Sucrose ester SP 70 [Sucrose stearate (CAS Number: 84066-95-5)] was a kind gift from Sisterna (Roosendaalc, The Netherlands). Cetostearyl alcohol [hexadecan-1-ol;octadecan-1-ol (CAS Number: 67762-27-0)], stearic acid [octadecanoic acid (CAS Number: 57-11-4)], propylene glycol [propane-1,2-diol (CAS Number: 57-55-6)], isopropyl myristate [propan-2-yl tetradecanoate (CAS Number: 110-27-0)], and Aknemycin™ ointment (20 mg/g erythromycin, Almirall Hermal, OGYI-T-2373/01) were purchased from Hungaropharma Ltd., (Budapest, Hungary). The MTT [2-(4,5-dimethyl-2-thiazolyl)-3,5-diphenyl-2H-tetrazolium bromide)] dye, Dulbecco’s Modified Eagle’s Medium (DMEM), phosphate buffered saline (PBS), trypsin from porcine, ethylene-diamine-tetra-acetic acid (EDTA), heat-inactivated fetal bovine serum (FBS), L-glutamine, 2,2-diphenyl-1-picrylhydrazyl (DPPH), absolute ethanol, ascorbic acid, (±)-6-Hydroxy-2,5,7,8-tetramethylchromane-2-carboxylic acid (Trolox) (CAS Number: 53188-07-1), Mueller–Hinton broth, and thioglycolate broth were purchased from Sigma-Aldrich (Budapest, Hungary). Nonessential amino acid solution and penicillin–streptomycin mix, GlutaMax™ supplement, 96-well plates, and cell culture flasks were obtained from Thermo-Fisher (Darmstadt, Germany). HaCaT cells (human keratinocyte cells) were obtained from Cell Lines Service (CLS, Heidelberg, Germany).

### 4.2. Preparation and Characterization of Dry Spirulina platensis Extract

The strains of cyanobacteria (*Spirulina platensis*) used in the study were bought from the Culture collection of Autotrophic Organisms (CCALA) of the Institute of Botany, encoded 026 Limnospira maxima (Setchell & N.L.Gardner) Nowicka-Krawczyk, Mühlsteinová & Hauer, which is equivalent to the *Arthrospira platensis* (strains SAG 85.79) of Sammlung von Algenkulturen Universität Göttingen (SAG) [26]. The cyanobacteria strains were grown in a batch culture of Zarrouk’s medium (used as standard in industrial production) at pH 8.2 in a box with a thermostat (20 ± 1 °C) with a photoperiod of 12 h light/dark [28,69]. Filaments were collected by centrifugation (J-10 rotor of Beckman Avanti J-25; 4500× *g*) after 12 days of cultivation, and lyophilized (Christ Alpha 1-2 LD plus). The lyophilized powder was used for further studies.

The phycocyanin concentration was determined according to Tarko et al. as the conditions of cultivation, and the strains of cyanobacteria were identical in our work. One hundred grams of dry matter of *Spirulina platensis* is equivalent to 461.4 ± 16.1 mg of phycocyanin [28].

### 4.3. Formulation of Spirulina platensis-Containing Creams

Different nonionic emulsifier agents were used for the formulations of P60, CR, TFS, and SP 70. The total emulsifier content was 3% in all oil-in-water creams. The mixture of cetostearyl alcohol, stearic acid, glycerol, and IPM was heated to 60 °C and homogenized. This formed the oily phase of the oil-in-water creams. The mixture of propylene glycol, emulsifier agent, and purified water was also heated to 60 °C, which formed the aqueous phase. The aqueous and oily phases were homogenized and cooled down to 25 °C. The final step was the addition of the active ingredient, the SPP, in suspended (compositions I–IV) or dissolved form. In the case of compositions V–VIII, 5.0 g of the SPP was dissolved in 14.2 g TC by Radelkis OP-912 magnetic stirrer (Radelkis, Budapest, Hungary) at room temperature for 3 h. The SPP was totally dissolved in TC.

### 4.4. Texture Analysis

Analysis of the textural properties was conducted in order to estimate the mechanical characteristics of compositions I–VIII. A compression test was carried out and the resistance of formulations was measured by a CT3 Texture Analyzer (Brookfield, Middleboro, MA, USA). The force exerted on the probe was recorded using Texture Pro CT Software (Brookfield Engineering Laboratories, MA, USA). The texture analyzer was equipped with a TA5 cylinder-type probe (12.7-mm diameter and 35-mm length) during the test. The trigger load (4.0 g), target (10.0 mm), and speed (0.50 mm/s) of the device were fixed. For the examinations, a jar filled with the given composition was positioned approximately 5.50 cm under the probe of the texture analyzer. Before measurement, the probe was lowered to the surface of the sample at a speed of 1 mm/s. After reaching the surface, the probe penetrated to a depth of 10.0 mm with a speed of 0.50 mm/s, and the force exerted on the probe was obtained. Both loading and unloading phases of the penetration curve were measured. The compression studies were performed at room temperature (25 °C). All measurements were done in quintuplicate. The average values and the standard deviation were calculated.

### 4.5. In Vitro Diffusion Studies

There are no standard methods for the investigation of drug diffusion from semi-solid dosage forms, as in solid formulations. In the US Food and Drug Administration (FDA) guidelines, the Franz vertical diffusion cell is stated as the most suitable device for investigating drug release and diffusion from topical formulations [38]. As its name implies, the first description of the diffusion cell was done by T. J. Franz [70].

A set of six Franz cells (Hanson Microette TM Topical and Transdermal Diffusion Cell System) was used for in vitro membrane diffusion and permeation studies. The receiving chamber was filled with 7.0 mL of 30% (*v*/*v*) alcohol as the receiving phase and was stirred with a magnetic rotor, which was set to 350 rpm. For diffusion experiments, cellulose-acetate synthetic membranes (pore size 0.45 µm) were used. In topical diffusion studies, it is advisable to make the membrane lipophilic to model the permeability of the stratum corneum. Therefore, the cellulose-acetate membranes were previously impregnated with IPM. A quantity of 1.0 g from each SPP-containing formulation (I–VIII) was weighed and placed on the membrane as the donor phase (equivalent to 0.2307 mg of phycocyanin). The cells were sealed with a glass disc and a retaining ring. The effective diffusion surface area of each membrane was 1.767 cm^2^. The receptor solution was kept at 32.0 °C with a thermostat for the entire experiment so that the temperature of the Franz cell membrane was like the physiological skin temperature. The diffusion study was performed for 105 min. Samples of 0.8 mL were collected from the receiving phase at predetermined time points of 15, 30, 45, 60, 75, 90, and 105 min using a special syringe and replaced with same quantity of fresh receiving phase. Samples were captured in a quartz cuvette and their absorbance was measured. The phycocyanin content of samples was measured at 620 nm using a UV spectrophotometer (Shimadzu, Tokyo, Japan) [10,71]. As a blank sample, 30% alcohol was used. A calibration curve was determined before the spectroscopic measurements of phycocyanin. A linear connection was found between the concentration of phycocyanin and the measured absorbance.

To compare the dissolution profiles of the different ointment compositions, the *f*_2_ similarity factor was calculated using MS Excel [29,72] (Equation (1)).
(1)$$f_{2}\;=\;50\;\times \;\log\left\{\;{\left[1+\left(1/n\right)\sum_{j=1}^{n}w_{j}{\left|R_{j}-T_{j}\right|}^{2}\;\right]}^{-0.5}\times 100\;\right\}$$
where *n* is the sampling number, *R_j_* and *T_j_* are the percentages of dissolved reference and test products at each time point *j*, and *w_j_* is an optional weight factor. The preparations are considered similar if the similarity factor (*f*_2_) is between 50 and 100 [29,72].

The phycocyanin release rate (k) was estimated from the slope of the amount of drug released per unit area (µg/cm^2^) versus the square root of time (min^½^). The diffusion coefficient (*D*) of the drug was determined from the drug concentration at a given t time (*Q*, µg/cm^2^), the initial concentration ($C_{0}^{\prime }$), and the diffusion time (*t*) (Equation (2)):(2)$$D=\frac{Q^{2}\times \;\pi }{{\left(2\left[C_{0}^{\prime }\right]\right)}^{2}\;\times \;t}$$

### 4.6. Cell Culturing

The HaCaT cell line was used in cell viability and antioxidant assays. Cells were grown in a plastic cell culture flask (Nunc™ EasyFlask™, Thermo-Fisher, Darmstadt, Germany) in Dulbecco’s Modified Eagle’s Medium, supplemented with 10% (*v*/*v*) heat-inactivated fetal bovine serum (FBS), 1% (*v*/*v*) non-essential amino acids solution, 4 mmol/L L-glutamine, 100 IU/mL penicillin, and 100 μg/mL streptomycin at 37 °C in an atmosphere of 5% CO_2_ [19].

The culture medium was changed twice per week. The cells were routinely maintained by regular passaging. The cells used for cytotoxic and antioxidant experiments were between passage numbers 20 and 40.

### 4.7. In Vitro Cell Viability Assay

For the cell viability assay test solutions of the compositions with and without SPP were prepared with the Franz diffusion cells. Each composition was investigated with three individual cells where PBS was used as the receptor phase instead of alcohol, thus eliminating false results due to the toxicity of the receptor medium. Test solutions of 1 mL were taken at the 105th minute, when the diffusion of phycocyanin reached its maximum.

The 0.50 (*m*/*m*) % solutions of SP70, TFS, P60, CR, TC, and SPP were prepared with PBS. Then, 0.50 g of the excipients was dissolved in 99.50 g PBS. The components were mixed by Radelkis OP-912 magnetic stirrer (Radelkis, Budapest, Hungary) at room temperature for 3 h in each case.

The cytotoxic effects of the compositions were evaluated using the MTT test following Mosmann [73]. The MTT test is a colorimetric assay based on the ability of the cells to metabolize. The activity of the nicotinamide adenine dinucleotide phosphate (NAD(P)H)-dependent cellular (mitochondrial) oxidoreductase (dehydrogenase) enzyme is examined, which converts water-soluble yellow MTT dye [3-(4,5-dimethylthiazol-2-yl)-2,5-diphenyl-tetrazolium bromide] into a water-insoluble dark-blue formazan crystal [(E, Z)-5-(4,5-dimethylthiazol-2-yl)-1,3-diphenylformazane] by opening the tetrazolium ring [74,75].

HaCaT cells in complete medium were seeded on 96-well plates at a final density of 10^3^ cells/well and allowed to grow in a CO_2_ incubator at 37 °C for four days. After that, the medium was removed, test solution was added, and the cells were incubated for 1 h with the samples. Then, the samples were removed, the cells were washed twice with 1 mL PBS, and 0.5 mg/mL MTT solution (MTT salt dissolved in PBS) was added to each well. The cells were incubated for a further 3 h, and then the MTT solution was removed and 0.1 mL of a solution of acidic isopropanol (isopropanol:1.0 M hydrochloric acid = 25:1) was added to each well to dissolve the formed, dark-blue formazan crystals. The absorbance was measured at 570 nm against a 690-nm reference with FLUOstar OPTIMA Microplate Reader (BMG LABTECH, Offenburg, Germany). The viability of the cells was expressed as the percentage of the cell viability of the untreated control cells, which were incubated with PBS for 1 h. The absorbance values correlated with the number of cells still alive. Each experiment was repeated three times with three wells for each sample.

### 4.8. Antioxidant Assay

#### 4.8.1. Superoxide Dismutase (SOD) Assay

For the antioxidant experiment, UV-B (Oriel^®^ Sol-UV-4 UV Solar Simulator, USA) radiation was used to induce oxidative stress, free radical formation, and cell damage before or after the treatment. The antioxidant activity of formulations III, IV, VII, and VIII was determined on HaCaT cells before and after UV-B exposure. Samples (1.0 mL) were collected using the Franz diffusion device, like in the case of the MTT assay (1.0 mL).

Samples (1.0 mL) were collected using the Franz diffusion device, like in the case of the MTT assay. As positive control, HaCaT cells were treated with Trolox, a water-soluble derivative of vitamin E. Trolox was dissolved in PBS immediately before use (10 µM).

The cells were seeded on 12-well plates at a density of 10^5^ cells/well and grown in a CO_2_ incubator at 37 °C for three days. In the pretreatment group, culture medium was removed, 10 µL of the test solutions were added, and the cells were incubated for a further 20 min with the samples. After 10 min of UV-B irradiation, samples were removed and cells were washed twice with PBS and incubated for 24 h. In the case of the posttreatment group, cells were irradiated for 10 min before the treatment, incubated with the test solutions for 20 min, washed with PBS, and then incubated for 24 h.

Cells (10^6^) were harvested using a rubber rod and centrifuged at 1000× *g* for 10 min at 4 °C. The cell pellet was then homogenized in 20 mM HEPES buffer (1 mM EGTA, 210 mM mannitol, and 70 mM sucrose/g tissue) and centrifuged at 10,000× *g* for 15 min at 4 °C at pH 7.2. The SOD activity of supernatant was analyzed using assay kits from Cayman according to the manufacturer’s instructions (Cat. 706002, Cayman, Ann Arbor, MI, USA). All experiments were performed in triplicate.

#### 4.8.2. DPPH Radical Scavenging Activity

DPPH (C_18_H_12_N_5_O_6_, M = 394.33 g/mol) radical scavenging activity of the formulations was measured using the method of Brand–Williams with minor modifications [76]. Samples from composition III, IV, VII, and VIII with or without SPP were collected using the Franz diffusion cells, as previously described. As receptor phase, absolute ethanol was used. The radical scavenging activity of *Spirulina platensis* extract (10.0 mg/mL ethanol solution) was also examined.

Then, 2.0 mL of DPPH radical solution (0.06 mM) in absolute ethanol was added to 900 µL of absolute ethanol. Finally, 100 µL of sample was added to the mixture and allowed to react at room temperature. The reaction mixtures were kept in the dark for 30 min to incubate. DPPH reacted with antioxidant compounds, which can donate hydrogen. When DPPH accepted hydrogen radical the reaction resulted in a color change from deep purple to light yellow [77]. Quantitative measurement of the remaining DPPH was carried out with UV-spectrophotometer (Shimadzu Spectrophotometer, Tokyo, Japan) at 517 nm [78]. In photometric determination, absolute ethanol served as background. Alcoholic solution of ascorbic acid (0.25 mg/mL) was used as standard in order to check the correctness of the measurement [79]. As negative control, 2.0 mL of DPPH solution (0.06 mM) diluted with 1.0 mL absolute ethanol was applied. The antioxidant activity percentage (AA% = antioxidant activity) was determined according to Mensor et al. [80] (Equation (3)):AA% = 100 − {[(Abs_sample_ − Abs_blank_) × 100]/Abs_control_}(3)

The EC_50_ value of *Spirulina platensis* extract and ascorbic acid, defined as the amount of antioxidant necessary to decrease the initial DPPH concentration (0.06 mM) by 50%, was also calculated from the results [79].

### 4.9. Antibacterial Test

The antibacterial effect of compositions VII and VIII was investigated on *S. aureus* (American Type Culture Collection (ATCC)^®^ 43300™) and *C. acnes* (ATCC^®^ 33169™) using a starting inoculum of 1 × 10^5^ cells/mL and 1 × 10^6^–10^7^ cells/mL, respectively. Antibacterial susceptibility tests were performed using the standard broth microdilution method in accordance with the recent European Committee for Antimicrobial Susceptibility Testing (EUCAST) protocols (determination of minimum inhibitory concentrations (MICs) of antibacterial agents by broth dilution (E.Dis) 5.1, determination of antimicrobial susceptibility test breakpoints (E.Def) 7.3.1) [81].

*S. aureus* was grown in Mueller–Hinton broth (MHB) under aerobic conditions at 37 °C overnight. Fifty microliters of *S. aureus* suspension were added to 5 mL of 2-fold concentrated MHB. *C. acnes* was grown in thioglycolate broth to support the growth of anaerobic microorganisms. It contained sodium thioglycolate, dextrose, sodium chloride, and nutritive factors, including casein, yeast, beef extract, and vitamins. The broth also contains 0.075% agar, which helps to create an anaerobic environment deeper in the tube [82]. The bacterial colony (clasper) was incubated for 24 h at 37 °C under anaerobic conditions in order to get a good culture of microorganisms.

Test solutions of compositions VII and VIII with and without SPP and Aknemycin™ ointment (20 mg/g erythromycin) were prepared with the Franz diffusion cells. Each composition was investigated with three individual cells where PBS was used as the receptor phase and the test solutions of 1 mL were taken at the 105th minute, when the diffusion of phycocyanin reached its maximum. Then, 10.0 mg of SPP was dissolved in 0.5 g TC, and then 3.5 g PBS was added to form the SPP test solution. The components were mixed by a magnetic stirrer at room temperature for 5 h. Negative control groups were treated only with PBS in both cases.

The broth microdilution assays were performed using 96-well plates. Then, 100 µL of the test solutions from each individual diffusion cells and of the SPP test solution were added to 100 µL of bacterial suspension on four separate wells. Plates were incubated for 6 h at 37 °C in a Concept 400 anaerobic chamber. During incubation, samples were taken after 2 and 6 h and their absorbance was measured at 600 nm by a Thermo-Fisher Multiskan Go (Thermo Fisher) microplate reader.

### 4.10. Statistical Analysis

Data were handled and analyzed using Microsoft Excel 2013 and GraphPad Prism (version 6; GraphPad Software, San Diego, CA, USA) and presented as means ± S.D. Comparison of multiple groups was performed with one-way ANOVA followed by either Dunnett’s multiple comparison test or Tukey’s multiple comparison test, depending upon whether the groups were compared to a given control group or to each other [83]. Results were regarded as significant at *p* < 0.05.

## 5. Conclusions

In our present study, topical *Spirulina platensis* formulations containing different surfactants were designed. Preparations that contained sucrose ester SP 70 emulsifying agent showed good dissolution profiles and the highest antioxidant activity against UVB-induced oxidative stress on HaCaT cells. Moreover, this formulation also had antimicrobial effect against *C. acnes* and *S. aureus* and caused low toxicity on HaCaT cells.

It can be concluded that cream containing *Spirulina platensis*, as a natural topical formulation, can be an alternative option to treat acne with fewer side effects and without antibiotic resistance.

## Figures and Tables

**Figure 1 molecules-25-04856-f001:**
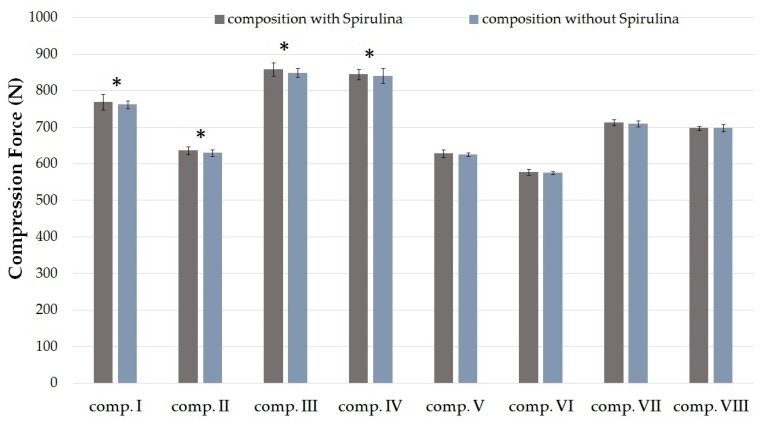
Texture analysis result of the SPP-containing compositions I–VIII (signified with gray columns) and compositions without SPP (signified with blue columns) at 25 °C, determined as compression force (N). Data are expressed as means ± S.D. and *n* = 5. Ordinary one-way ANOVA and Tukey’s multiple comparison test were performed to compare formulations with and without SPP and suspended and dissolved formulations. Significant differences are marked on the figure with * (*p* < 0.05). The application of Spirulina platensis powder did not statistically change the compression force values in any of the cases. Significant differences were detected between the suspended (I–IV) and the dissolved forms (V–VIII) in every pair of compositions.

**Figure 2 molecules-25-04856-f002:**
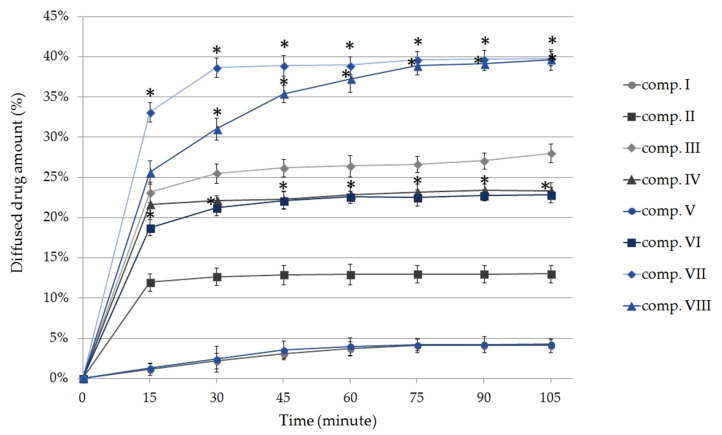
The in vitro membrane diffusion profiles of phycocyanin through isopropyl myristate (IPM) impregnated cellulose-acetate synthetic membrane from formulations I–VIII. Bars represent the mean ± S.D. of six experiments. Ordinary one-way ANOVA and Tukey’s multiple comparison test were performed to compare suspended and dissolved formulations. Statistically significant differences are marked with asterisks at *p* < 0.05. The * shows the significant differences in the case of compositions II and VI, III and VII, and IV and VIII (containing SPP in suspended or dissolved forms) examined at the same times.

**Figure 3 molecules-25-04856-f003:**
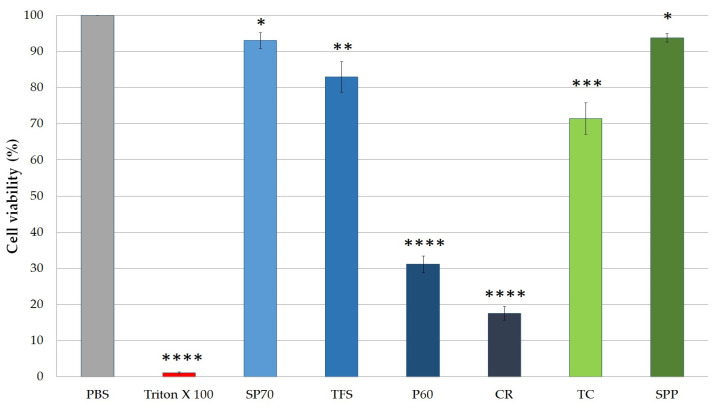
Cell viability evaluation following MTT assay on HaCaT cells treated with 0.50 (m/m)% solutions of sucrose ester SP 70 (SP70), Tefose 63 (TFS), Polysorbate 60 (P60), Cremophor A6 and A25 (CR), Transcutol HP (TC), and *Spirulina platensis* powder (SPP). Each data point represents the mean ± S.D. and *n* = 10. Cell viability is expressed as the percentage of negative control (PBS), which was treated only with PBS. The positive control was Triton X 100 (10% *w*/*v*), which had significantly lower cell viability results compared with the untreated control. Ordinary one-way ANOVA with Dunnett’s multiple comparison test was performed to compare the different solutions with PBS. The *, **, ***, and **** indicate statistically significant differences at *p* < 0.05, *p* < 0.01, *p* < 0.001, and *p* < 0.0001.

**Figure 4 molecules-25-04856-f004:**
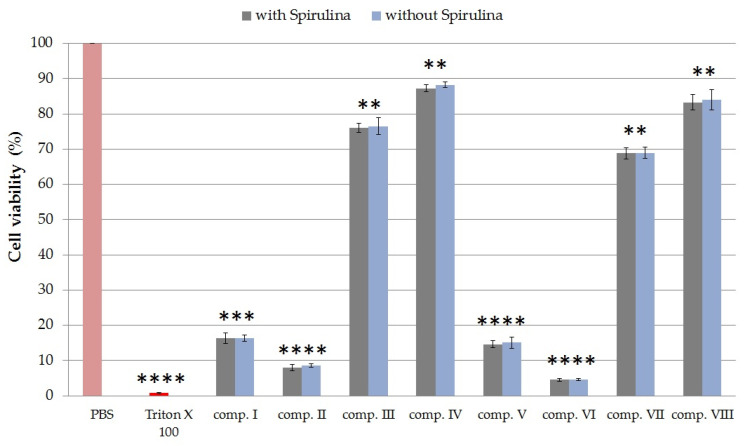
Cell viability test with the MTT assay on HaCaT cells after incubation of compositions I–VIII for 1 h. Cell viability is expressed as the percentage of negative control (PBS), which was treated only with PBS. The positive control was Triton X 100 (10% *w*/*v*), which had significantly lower cell viability results compared with the untreated control. Each data point represents the mean ± S.D. and *n* = 12. Ordinary one-way ANOVA with Dunnett’s multiple comparison test was performed to compare the different formulations with PBS. The application of Spirulina platensis powder did not statistically change the cell viability values in any of the cases. The **, ***, and **** indicate statistically significant differences at *p* < 0.01, *p* < 0.001, and *p* < 0.0001.

**Figure 5 molecules-25-04856-f005:**
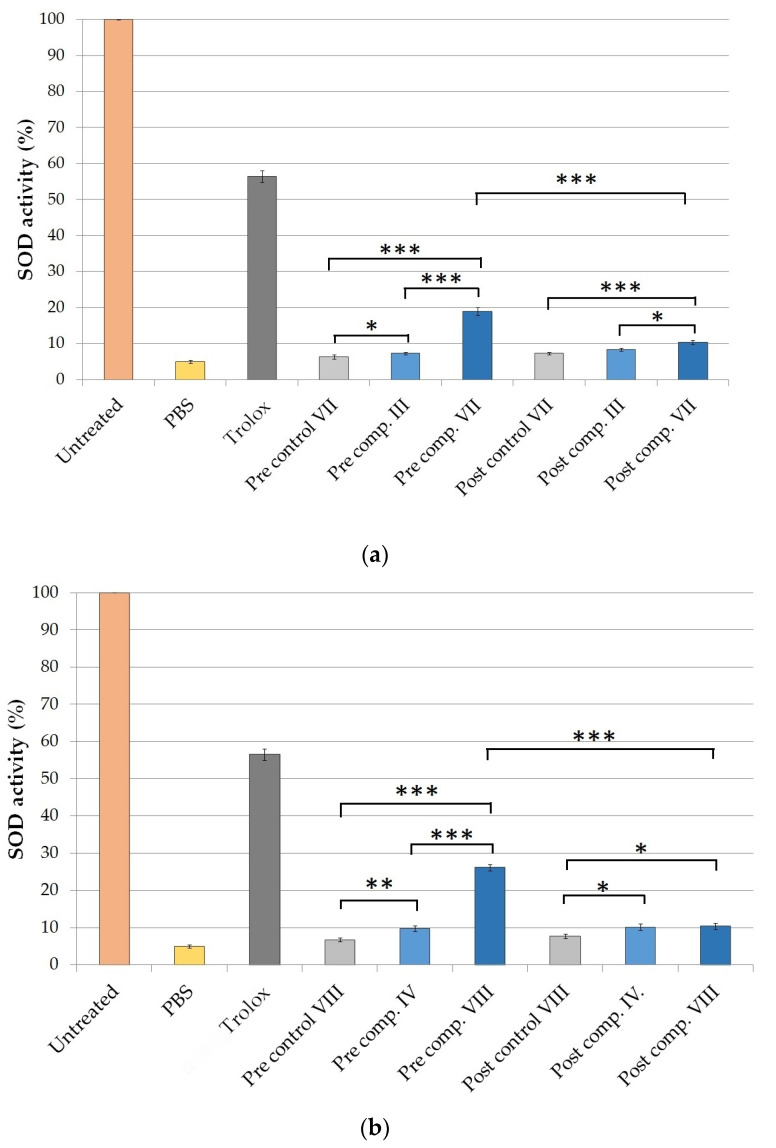
Effects of pre- and posttreatment with compositions containing Tefose 63 (comp. III, VII) (**a**) and sucrose ester SP 70 surfactant (comp. IV, VIII) (**b**) on superoxide dismutase (SOD) enzyme activity in HaCaT cells exposed to UVB irradiation. SOD activity is expressed as the percentage of SOD activity in untreated keratinocytes. UVB irradiated cells pretreated with PBS were used as negative control. As positive control, Trolox (Vitamin E derivate) (10.0 µM) dissolved in PBS was used in pretreatment. Data are expressed as the mean ± S.D. and *n* = 10. Ordinary one-way ANOVA and Tukey’s multiple comparison tests were performed to compare the formulations with each other. The *, **, and *** indicate statistically significant differences at *p* < 0.05, *p* < 0.01, and *p* < 0.001.

**Figure 6 molecules-25-04856-f006:**
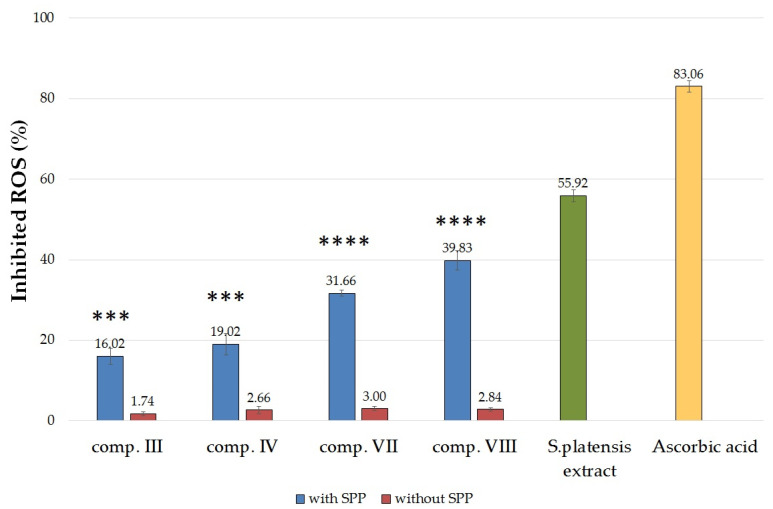
DPPH-scavenging activity of SPP-containing compositions compared with formulations without SPP. The positive control was S. platensis extract in a concentration 10 mg/mL and ascorbic (0.25 mg/mL) dissolved in ethanol (96%). As negative control, 2.0 mL of DPPH solution (0.06 mM) diluted with 1.0 mL absolute ethanol was applied. Data presented as mean ± SD (*n* = 4). Ordinary one-way ANOVA and Tukey’s multiple comparison test were performed to compare formulations with or without SPP. Significant differences are marked on the figure with *** and **** (*p* < 0.001 and *p* < 0.0001), showing the significance levels in the case of compositions containing SPP and the same composition without SPP. Significant differences were detected in every case between the formulations with SPP and without SPP.

**Figure 7 molecules-25-04856-f007:**
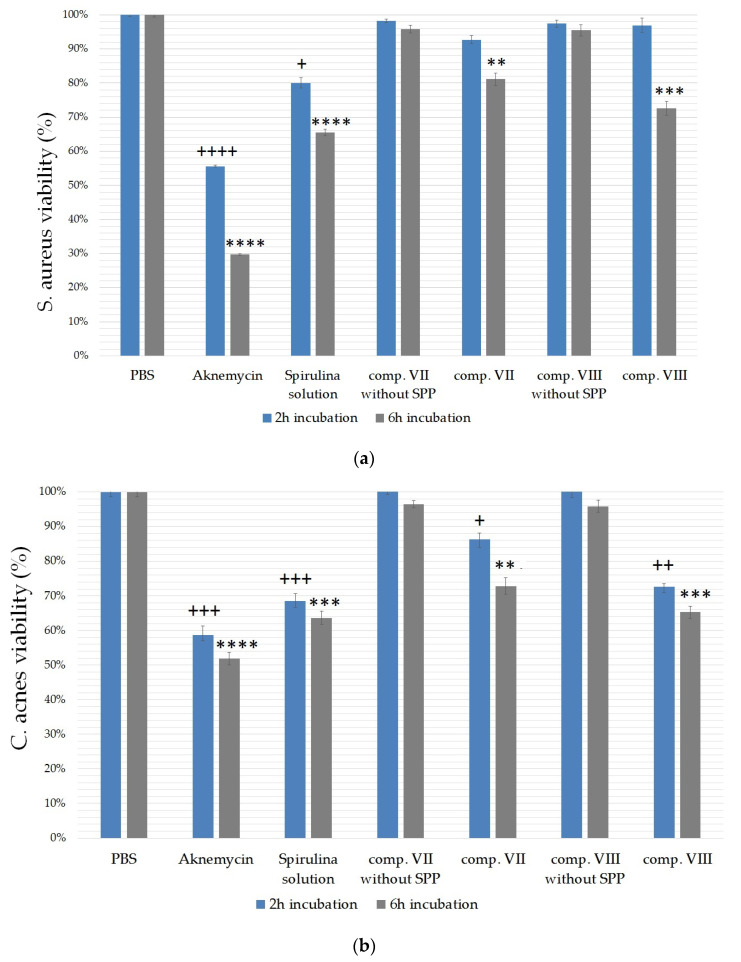
Cell viability of *S. aureus* (**a**) and *C. acnes* (**b**) against the compositions VII and VIII, the positive control (Aknemycin™), and the SPP solution (0.25%). Cell viability is expressed as the percentage of the absorbance of the negative control group, which was treated only with PBS. Data are expressed as mean ± SD and *n* = 12. Statistical significance is indicated by +, ++, +++, ++++, and **, ***, **** at *p* < 0.01, *p* < 0.001, and *p* < 0.0001 in the cases of the 2-h and 6-h samples, respectively. Treated groups were compared to PBS with one-way ANOVA followed by Dunnett’s multiple comparison test.

**Table 1 molecules-25-04856-t001:** Compositions of *Spirulina platensis*-containing creams.

Composition	SPP (5 g)	TC (14.2 g)	Nonionic Emulgents				Cetostearyl Alcohol (4.6 g) Stearic Acid (10 g) Glycerol (5 g) IPM (5 g) Propylene Glycol (5 g) Purified Water (ad 100 g)
			P60 (3 g)	CR (3 g)	TFS (3 g)	SP70 (3 g)	
I	+		+				+
II	+			+			+
III	+				+		+
IV	+					+	+
V	+	+	+				+
VI	+	+		+			+
VII	+	+			+		+
VIII	+	+				+	+

Abbreviations: SPP (*Spirulina platensis* powder), TC (Transcutol HP), P 60 (Polysorbate 60), CR (Cremophor A6 and A25 in the ratio 1:1), TFS (Tefose 63), SP70 (sucrose stearate SP 70), IPM (isopropyl myristate).

**Table 2 molecules-25-04856-t002:** The pH values of the final formulations. Each data point represents the mean ± S.D. and *n* = 5.

Composition	pH Value ± SD
I	6.77 ± 0.04
II	6.83 ± 0.04
III	6.95 ± 0.05
IV	6.92 ± 0.02
V	6.77 ± 0.04
VI	6.82 ± 0.02
VII	6.90 ± 0.03
VIII	6.91 ± 0.03

**Table 3 molecules-25-04856-t003:** The comparison of dissolution profiles of the dissolved and suspended forms (containing the same emulsifier) by the calculation of the similarity factor (*f*_2_).

Pairwise Comparison	*f* _2_
comp. VII vs. comp. III	45.42
comp. VIII vs. comp. IV	43.59
comp. VI vs. comp. II	53.91
comp. V vs. comp. I	97.74

**Table 4 molecules-25-04856-t004:** Phycocyanin release rate and the diffusion coefficient values related to the compositions (I–VIII). Each data point represents the mean ± S.D., *n* = 6, and *p* < 0.05. Ordinary one-way ANOVA and Tukey’s multiple comparison test were performed to compare suspended and dissolved formulations. Significant differences are marked with * in the table. Asterisks show the significance levels in the case of compositions containing SPP in suspended or dissolved forms.

Composition	Release Rate	Diffusion Coefficient
	k 10^2^ (µg/cm^2^ min^1/2^) ± S.D.	D 10^5^ (cm^2^/min) ± S.D.
I	59.12 ± 2.08	0.0178 ± 0.002
II	144.4 ± 6.21	0.1777 ± 0.015
III	313.01 ± 11.3	0.8227 ± 0.041
IV	258.09 ± 5.96	0.5699 ± 0.032
V	60.72 ± 4.51	0.0191 ± 0.002
VI	263.34 ± 11.63	0.5470 ± 0.028 *
VII	455.41 ± 22.62	1.6601 ± 0.091 *
VIII	490.42 ± 27.87	1.6472 ± 0.083 *

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
