# Peer review of "Formulation of Creams Containing Spirulina Platensis Powder with Different Nonionic Surfactants for the Treatment of Acne Vulgaris"

_molecules, 2020, doi:10.3390/molecules25204856_

Round 1

Reviewer 1 Report

The authors responded to the objections raised in the study by introducing missing information. IUPAC names, C.A.S. number were specified, the pH of each emulsion was detected.

However, the pH of the emulsions is higher than that of the skin, where it rarely approaches neutrality based on anatomical area. Why choosing a neutral pH? The study shows that the experiments were conducted using a preservative-free emulsion highlighting the anti-acne activity of the Spirulina platensis powder  in vitro compatible with a preliminary study of a topical formulation

Author Response

Thank You for reviewing the manuscript.

Reviewer 2 Report

Józsa L. et al. present a manuscript describing the formulation of different creams containing spirulina with nonionic surfactants for the treatment of acne vulgaris.  This work represents a new submission of a manuscript previously reviewed. The new version of the manuscript features improvements and the comments previously raised have been adequately addressed by the authors.

I have an important question for the authors regarding the statistical analysis:

-While the ANOVA test is appropriate for the analysis of the means relative to the different samples compared with each other, the analysis of the Pearson correlation coefficient (r) does not seem correct. If the authors intended to report the analysis of Pearson's correlation, it would be necessary to indicate the variables towards which the analysis was carried out and the corresponding r value.

- Why in Figure 1 was the statistical analysis performed only for SPP-containing compositions I – VIII signed with gray columns?

Author Response

Thank You for reviewing manuscript

Reviewer 3 Report

After reviewing the new version of the manuscript, I agree with its content and consider that it can be published.

Author Response

Thank You for reviewing manuscript.

This manuscript is a resubmission of an earlier submission. The following is a list of the peer review reports and author responses from that submission.

Round 1

Reviewer 1 Report

The authors describe preparation of eight cream formulations, all containing the same amount of a cyanobacteria preparation mostly called spirulina. The authors have analyzed the antibacterial and antioxidant activities of the creams in vitro. There are some serious shortcomings of the manuscript. Thus, the conclusion on favourable effect of creams on pathogenic bacteria or oxidative parameters cannot be confirmed as the authors did not compare their formulations to those containing the same surfactants but no spirulina. In addition, all but one experiment show no positive control. Moreover, an undefined “preservative solution” was used in the formulations (p. 11, line 351). Its effect on the parameters analyzed should have been tested.
Penetration and cytotoxicity studies should be evaluated in vivo.
Figures represent data just from multiple measurements of the same experiment that cannot be analyzed statistically. The authors should show results from at least three different cream preparations. The quantitative data might be not precise as assay conditions were not precisely controlled: “The receiving chamber was filled with approximately 7.0 mL of …”
Abstract is less informative. It should contain information on formulations used and results obtained. Methods should not be simply listed but used to illustrate results and support their validity. Abstract should contain a definition of what spirulina is; the Latin name of the representative bacteria should be given.
Discussion should contain interpretation of the results in the context of the available knowledge. Results should be moved from Discussion to Results. In general, Discussion is not informative.
Toxicity analysis and antioxidant assay. The legends should contain the information that just unknown compounds, which penetrated through the membrane of Franz diffusion cell (exact time should be given) from either cream formulation into PBS have been tested for cytotoxicity or antioxidant effect against HaCaT cells, as well as, that the treatment time was just 20-30 min. Longer exposure times and culture medium instead of PBS should be used to analyze possible toxic effects on keratinocytes. In vivo effects on mice skin with subsequent immunohistochemistry would be here more informative.
Fig. 1,2 and also 3 ; Table 3. Figures show some statistical differences among 8 formulations prepared. However, it is not clear between which groups the difference exists. Particularly important, as in Fig. 1-2 and Table 3 there are no respective negative (without spirulina) and no positive controls are shown.
P. 5. Before concluding that “VI and II can also be considered similar, since the f2 value (53.91) is between 50 and 100”, the authors should define criteria when two preparations are considered similar.
Page 2: What TC means? “To enhance the permeability of Spirulina and to improve its solubility TC was also added to the formulations”.
P.4, lines 128-143. It should be given which treatment time was used for these evaluations as kinetics of phycocyanin diffusion is different between creams.
It should be shown that the other components from the creams do not interefere with phytocyanin measurement (diffusion assay).
For the analysis of dissolution profiles, it should be given how Rj and Tj, the percent dissolved of the reference and the test products at each time point j, were calculated. What was the reference? How wj, an optional weight factor, was defined?
Poorly defined term spirulina should be avoided and the Latin name of the strain should be used throughout the text.

Reviewer 2 Report

The manuscript is well done, however it seems to me that it could be improved before being published based on the following comment:

As we know, a wide variety of metabolites are present in natural products and spirulina is no exception. Although in the manuscript (lines 299-301) they mention that both acetone and aqueous extracts have shown antimicrobial activity against S. aureus and C. acnes respectively, it would be useful to demonstrate which of all the metabolites present in the natural product are responsible for this biological activity, or if it is a synergistic effect.

Reviewer 3 Report

Józsa L. et al. present a manuscript describing the formulation of different creams containing spirulina with nonionic surfactants for the treatment of acne vulgaris. The study design and the interpretations of the results are reasonable.

My only objection concerns the way in which the authors determined the effect of spirulina on the activity (level) of SOD. They measure the antioxidant action of spirulina on HaCaT cells treated with creams and subsequently subjected to UV radiation. Why do UVB-treated cells have such a drastic reduction in SOD activity (about 95%) when compared to untreated cells? Is it a consequence of the deleterious action of UV on the DNA or of ROS? What is the viability of the cells at the end of the UV treatment? The authors claim that dissolved spirulina in TC (VII, VIII) with Tefose and SP 70 surfactant caused a more significant increase in the SOD level. It would be more correct to say that they have the lowest reduction in SOD activity as a consequence of UV radiation. Could it be the SOD activity dependent on the SOD present in cream residues? Why didn't the authors directly measure the ROS created as a result of UV treatment?

Minor considerations

- Explain in the abstract the purpose of the determination of the SOD;

- Line 39: “With pilosebaceous units”: please correct “involving pilosebaceus units”;

- Line 41: “and the products of inflammation”: “and inflammation”;

- Line 42: “C. acnes and S. aureus bacteria”: “Cutibacterium acnes (C. acnes) and Staphylococcus aureus (S. aureus);

- Lines 53-54 : the correct scientific names are Arthrospira (genus) and Arthrospira maxina (A. maxima) and Arthrospira platensis (A. platensis), whereas the term spirulina is more commonly understood as the commercial preparation. Please correct it adequately;

- Line 76: “Nonionic amphiphil surfactants: “Nonionic amphiphilic surfactants”;

- Line 104: it is not clear what the authors mean by "in both cases";

- Line 134: it is not clear what the authors mean by "in both cases";

- Line 153: as for the f2 factor, it would be useful for readers the formulas for the diffusion coefficient D and the release rate K;

- Line 166 paragraph 2.3:  As reported in Materials and Methods, MTT test were carried out using samples taken from the Franz cells. Please specify this procedural aspect while describing the results;

- Line 183 paragraph 2.4, Line 429 paragraph 4.8: It is not clear what the authors used to determine the antioxidant activity of spirulin compositions: did the authors use 10 ul of cream or of the respective sample taken from Franz's cells? If the cream was used, how the cream was evenly distributed over the surface of the well?

- Figure 5B: please correct P. acnes on the ordinate axis;

- Materials and methods: Please indicate how spirulin was dissolved in TC;

- Please report the names of the species and the Latin phrase “in vivo” in italics throughout the manuscript.

Reviewer 4 Report

The study is very interesting and provides a novelty from an application point of view of spirulina algae. However, there are aspects to be improved regarding the information provided in the report and in the experimental activity.

It would be advisable to insert the excipients of the formulation (emulsifiers, oils, emollients etc) with the official IUPAC nomenclature, where possible, or current denomination and C.A.S. number.

In the various sections of the work the commercial name is indicated always.

The cosmetic formulation does not indicate the preservatives neither as a commercial name nor in composition.

The cosmetic formulation does not indicate the oils used in composition

The pH of the final formulation is not indicated.

Bacterial culture tests compare the cosmetic formulation vs. pharmacological antibiotic ointments, but the activity of the preservative in the cosmetic formulation is neglected, that is, it would be useful to exclude the contribution of the preservative as a functional active substance in acne.

In order to understand the potential antibacterial activity of spirulina alga, it would be useful to carry out experiments with the formulation free of preservatives, to better characterize the possible anti-acne function of spirulina alga.